# Functional Analyses of a *Rhodobium marinum* RH-AZ Genome and Its Application for Promoting the Growth of Rice Under Saline Stress

**DOI:** 10.3390/plants14162516

**Published:** 2025-08-13

**Authors:** Yang Gao, Cheng Xu, Tao Tang, Xiao Xie, Renyan Huang, Youlun Xiao, Xiaobin Shi, Huiying Hu, Yong Liu, Jing Peng, Deyong Zhang

**Affiliations:** 1Yuelushan Laboratory, Institute of Microbiology, Hunan Academy of Agricultural Sciences, Changsha 410125, China; gaoyang@hunaas.cn; 2Yuelushan Laboratory, Institute of Plant Protection, Hunan Academy of Agricultural Sciences, Changsha 410125, China; xxxcheng@stu.hunau.edu.cn (C.X.); tangtao@hunaas.cn (T.T.); xiewei@hunau.edu.cn (X.X.); huangry0106@hunaas.cn (R.H.); xiaoyoulun@hunaas.cn (Y.X.); shixiaobin@hunaas.cn (X.S.); waddles1021@163.com (H.H.); liuyong@hunaas.cn (Y.L.); 3College of Plant Protection, Hunan Agricultural University, Changsha 410128, China

**Keywords:** *Rhodobium marinum*, genomic analysis, salt stress, plant growth promotion, biocontrol agent

## Abstract

Soil salinity stands among the most critical abiotic stressors, imposing severe limitations on global rice cultivation. Emerging evidence highlights the potential of beneficial microorganisms to enhance crop salt tolerance. In this study, a halotolerant bacterial strain, *Rhodobium marinum* RH-AZ (Gram-negative) was identified and analyzed. It exhibited exceptional survival at 9% (*w*/*v*) NaCl salinity. Whole-genome sequencing revealed a circular chromosome spanning 3,875,470 bp with 63.11% GC content, encoding 5534 protein-coding genes. AntiSMASH analysis predicted eight secondary metabolite biosynthetic gene clusters. Genomic annotation identified functional genes associated with nitrogen cycle coordination, phytohormone biosynthesis, micronutrient management and osmoprotection. Integrating genomic evidence with the existing literature suggests RH-AZ’s potential for enhancing rice salt tolerance and promoting the growth of rice plants. Subsequent physiological investigations revealed that the RH-AZ strain had significant growth-promoting effects on rice under high salinity stress. Compared with a non-inoculated control, RH-AZ-inoculated rice plants exhibited stem elongation and fresh biomass enhancement under salt stress conditions. The RH-AZ strain concurrently affected key stress mitigation biomarkers: it enhanced the activity of antioxidant enzymes including superoxide dismutase, peroxidase, catalase and ascorbate peroxidase, and the contents of proline and chlorophyll in plants, and reduced the content of malondialdehyde. These findings demonstrate that *R. marinum* RH-AZ, as a multifunctional bioinoculant, enhances rice salt tolerance by enhancing the stress responses of the plants, presenting a promising solution for sustainable agriculture in saline-affected ecosystems.

## 1. Introduction

Over 20% of the total agricultural land worldwide has been affected by soil salinity, which represents one of the most devastating factors that cause the deterioration of arable land, and the proportion will increase to 50% by 2050 [1,2]. Rice (*Oryza sativa* L.), a staple crop for over half the world’s population [3], is particularly sensitive to salt stress, which disrupts ion homeostasis, suppresses growth, and significantly diminishes grain yields under extremely high saline concentrations [4]. With the global population projected to exceed 10 billion by 2050 according to the *2020 Global Agricultural Productivity (GAP) Report*, developing sustainable strategies to enhance rice salinity tolerance has become critical for food security.

The utilization of plant growth-promoting rhizobacteria (PGPR) under adverse environmental conditions represents an environmentally friendly strategy to enhance crop yield [5,6]. However, high osmotic stress and ionic toxicity (e.g., from Na^+^ and Cl^−^) induced by soil salinization threaten microbial survival in the soil ecosystem [7]. Consequently, halotolerant microorganisms derived from high-salinity environments such as marine systems have become a focal point in soil salinization research. Studies have demonstrated that halotolerant PGPR can improve seed germination rates under salt stress [8,9]. Inoculating crops with these salt-tolerant PGPR strains effectively enhances crop salt tolerance [10,11,12]. This evidence highlights the potential of halotolerant PGPR to alleviate salt stress through mechanisms including osmoregulation, antioxidant defense, and phytohormone modulation [13,14]. Microbially mediated plant salt tolerance has been proved in diverse plant species, such as *Arabidopsis thaliana*, *Oryza sativa* L., *Brassica campestris*, *Triticum aestivum* and *Zea mays* [1,2,8,15,16,17].

Photosynthetic bacteria (PSB) are a diverse group of microorganisms capable of photoautotrophy through bacteriochlorophyll-mediated processes. PSB have achieved significant attention as a sustainable resource for developing microbial pesticides, owing to their environmental compatibility and plant growth-promoting properties, which enhance crop agronomic traits [18,19]. Inoculation studies with *Rhodopseudomonas palustris*, either as a single strain or in combination with other strains, demonstrated remarkable improvements in rice cultivated in saline–alkali soils. Key parameters such as plant height, root length, dry biomass, chlorophyll content, panicle weight, grain number per panicle, and yield exhibited significant increases or consistence with control groups [20,21]. While PSB show immense potential as a microbial bioagent for improving crop salt tolerance, their molecular mechanisms still remain unclear.

Although traditional breeding and genetic engineering approaches have achieved some success in developing salt-tolerant crops, these methods remain time-consuming, technically complex, and costly [22]. Utilizing beneficial microorganisms to enhance crop salinity tolerance offers a cost-effective and environmentally sustainable alternative. This study identified and characterized the *Rhodobium marinum* strain RH-AZ, a highly halotolerant phototrophic bacterium. Inoculation experiments using salt-sensitive (cultivar 9311) and moderately salt-tolerant (cultivar 3931) rice demonstrated RH-AZ’s ability to significantly promote growth under saline stress. Whole-genome sequencing and analysis of RH-AZ were performed, with secondary metabolite biosynthetic gene clusters predicted using AntiSMASH. Genome annotation facilitated functional analysis of salinity mitigation-related genes to assess RH-AZ’s potential in alleviating salt stress. We further investigated RH-AZ’s role in improving rice growth performance under salt stress. This study will help elucidate the physiological mechanisms by which RH-AZ enhances salt tolerance in rice, supporting the development of sustainable bio-strategies for coastal agriculture and saline-affected soils.

## 2. Results

### 2.1. Characterization of RH-AZ

An isolated bacterial strain, identified through 16S rDNA sequence analysis, was classified as *Rhodobium marinum,* namely, RH-AZ (GenBank accession no. PP711544.1). The RH-AZ strain is a Gram-negative bacterium within the class *Alphaproteobacteria*, order *Rhizobiales*, family *Rhodobiaceae*, and genus *Rhodobium*. Colonies grown on double-layer agar plates exhibited a dark-red pigmentation with circular morphology; phototrophic liquid cultures developed a rose-red coloration (Figure 1A), while the strain’s fermentation supernatant appeared light yellow (Figure 1B). Absorption spectra of RH-AZ cell suspensions showed characteristic bacteriochlorophyll peaks at 830 and 890 nm. When cultured in nutrient broth under optimal conditions (30–35 °C, pH = 7.0–7.5, 3–6% (*w*/*v*) NaCl), the RH-AZ strain reached the mid-exponential growth phase within 4–5 days, achieving a colony density of 10^8^ Colony-Forming Units per milliliter (CFU·mL^−1^). *Rhodobium* species typically inhabit marine and hypersaline environments. The RH-AZ strain demonstrated halotolerance across a salinity gradient of 0–10% NaCl, with optimal growth at 3–6% NaCl and no observable growth in NaCl-free medium. The RH-AZ strain exhibited normal metabolic activity in media containing up to 9% NaCl (Figure 1C), confirming its adaptation to hypersaline conditions. Scanning electron microscopy (SEM) revealed short, rod-shaped cells (0.8–1.2 μm in length) with obtuse ends (Figure 1D). Transmission electron microscopy (TEM) revealed the internal structures such as the photosynthetic layers and other structures of the stain (Figure 1E).

### 2.2. Phylogenetic Classification of RH-AZ

Phylogenetic analysis based on 16S rDNA sequences involved 17 nucleotide sequences. All ambiguous positions were removed for each sequence pair (pairwise deletion option). There were a total of 1544 positions in the final dataset. The analysis result demonstrated that RH-AZ exhibits 93% sequence identity with two documented *R. marinum* strains (GenBank accession nos. EU445270.1 and AM696708.1), while showing significantly lower similarity (<85%) to other reference strains. This comparative genomic evidence conclusively establishes RH-AZ’s phylogenetic affiliation within the *R. marinum* clade (Figure 2).

### 2.3. Whole-Genome Sequencing and Functional Annotation of RH-AZ

The complete genome of RH-AZ comprises a single circular chromosome spanning 3.88 Mb (GC content: 63.11%) devoid of plasmids (Figure 3A). Genomic architecture analysis revealed that 49 tRNAs, 6 rRNA operons (5S-16S-23S: 2-2-2), and 24 noncoding RNAs were predicted in the genome. Two kinds of repetitive elements—149 tandem repeats (TRFs) and 15 simple sequence repeats (SSRs)—were identified in the genome. Additionally, three kinds of mobile genetic elements were found: three prophage regions, four genomic islands (GIs), and two CRISPR-Cas arrays.

Functional annotation assigned roles to 3761 protein-coding sequences (97.96% genome coverage), with 3518 genes (93.5%) classified into 25 Clusters of Orthologous Groups (COG) categories (Figure 3B). Key functional enrichments included the following: amino acid metabolism (COG-E, 11.14%), general function prediction only (COG-R, 7.56%), carbohydrate utilization (COG-G, 6.59%), coenzyme transport and metabolism (COG-H, 6.59%), transcriptional regulation (COG-K, 6.54%), inorganic ion transport and metabolism (COG-P, 6.54%), ribosomal biogenesis (COG-J, 6.28%), and cell envelope dynamics (COG-M, 6.14%).

### 2.4. Genes Analysis of Plant Growth Promotion and Salt Stress Tolerance

Genome mining revealed the presence of functional genes associated with both plant growth promotion and abiotic stress alleviation. These genes included those involved in nitrogen fixation, nitrogen metabolism, auxin biosynthesis, acetolactate synthase, siderophore transporter, phosphorus solubilization, oxidative stress alleviation, K^+^/Na^+^ transporter, and osmoprotectant (glycine betaine, trehalose, and proline and choline) synthesis. The specific pathways of these genes that participated in, along with their locus tags, gene names, and coding gene products, are summarized in Appendix A. Among these, 25 genes were related to nitrogen fixation, 16 genes to nitrogen metabolism, 8 genes to auxin biosynthesis, 4 genes to acetolactate synthase, 12 genes to siderophores transport, 14 genes to phosphorus solubilization, 13 genes to oxidative stress alleviation, 13 genes to K^+^/Na^+^ transporter, and 41 genes to osmoprotectants (glycine, betaine, and proline) synthesis. Additionally, secondary metabolite biosynthesis was predicted via antiSMASH analysis, revealing eight specialized clusters in the genome of the RH-AZ strain, including two terpenes, three hydrogen-cyanide, one N-acetylglutaminylglutamine amide (NAGGN), one ribosomally synthesized and post-translationally modified peptide-like (RiPP-like), and one type III polyketide synthase (T3PKS). These clusters were identified as secondary metabolites (Appendix A).

### 2.5. Effects of Salt Stress and Bacterial Inoculation on Plant Growth Parameters

Salt stress and bacterial inoculation exerted distinct morphological effects on both rice cultivars 3931 (moderately salt-tolerant) and 9311 (salt-sensitive) (Figure 4). Greater shoot elongation in the moderately salt-tolerant cultivar 3931 compared to the salt-sensitive cultivar 9311 (*p* < 0.05) was observed across all treatments, demonstrating genotype-specific responses to experimental conditions. Quantitative analysis of both rice cultivars revealed significant growth inhibition in salt-stressed plants in T2, T3, and T6 treatments relative to untreated control (T1), with reductions of 7.3–12.5% in stem length, 16.5–18.7% in stem fresh weight, and 15.7–23.3% in root fresh weight. The difference in root length, stem dry weight, and root dry weight were of low significance or no significance at all (Figure 5 and Figure 6). In contrast, bacterized plants in T4 and T5 treatments maintained growth metrics comparable to controls (Δ < 5%, *p* > 0.05), indicating effective stress mitigation by microbial inoculation. After a 2-week recovery period post-stress, rice plants in T4 and T5 treatments basically achieved phenotypic recovery, showing no significant differences compared to the T1 control. Salt-stressed plants in T2, T3, and T6 treatments exhibited persistent chlorosis and significant height reduction compared to the T1 control, demonstrating irreversible stress damage.

### 2.6. Effects of Salt Stress and Bacterial Inoculation on Chlorophyll Content, Malondialdehyde (MDA), and Proline (Pro)

The photosynthetic pigment analysis revealed genotype- and treatment-specific responses in moderately salt-tolerant cultivar 3931 and salt-sensitive cultivar 9311 (Figure 7A). Untreated controls (T1) maintained maximal chlorophyll content, while salt-stressed plants in T2, T3, and T6 treatments showed a marked reduction with 23% (*p* < 0.05). Bacterized plants in T4 and T5 treatments maintained chlorophyll levels comparable to T1 control (Δ < 5%, *p* > 0.05), with salt-sensitive cultivar 9311 exhibited marginally lower values than moderately salt-tolerant cultivar 3931 across all treatments.

Lipid peroxidation, quantified via MDA accumulation, was increased by 2.28–3.12-fold (*p* < 0.05) in salt-stressed plants in T2, T3, and T6 treatments relative to the T1 control, peaking in the salt-sensitive cultivar 9311 (Figure 7B). Microbial inoculation suppressed MDA elevation in the T4 and T5 treatment plants to 14–76% of baseline levels, with consistent mitigation observed at 3-, 6-, and 9-day sampling intervals post-stress induction.

Proline dynamics displayed contrasting patterns (Figure 7C). Salt stress triggered a 1.5–2.8-fold (*p* < 0.05) increase in proline accumulation in T2, T3, and T6 treatment plants, disproportionately higher in salt-sensitive cultivar 9311. Relative to T1 controls, bacterial treatment (T4/T5) significantly elevated proline synthesis by 2.0–3.6-fold (*p* < 0.05), exceeding Pro accumulation under solitary salt stress. This potentiation of osmoprotectant biosynthesis implies microbially facilitated metabolic reprogramming.

### 2.7. Effects of Salt Stress and Bacteria Inoculation on Antioxidant Enzyme Activity

Across both rice cultivars, salt stress induced significantly upregulated activities of key antioxidant enzymes including peroxidase (POD), catalase (CAT), superoxide dismutase (SOD), and ascorbate peroxidase (APX), which were 1.3–1.9-fold (*p* < 0.05) higher relative to the unstressed control of T1 (Figure 8). Microbial inoculations including T4 and T5 further potentiated this response, increasing by 1.5–2.7-fold compared to the control group, and with a 16–34% increase over salt-stressed treatments of T2, T3, and T6.

## 3. Discussion

Soil salinity poses a significant threat to rice cultivation. To address the urgent need for environmentally sustainable agricultural practices, biological control methods, particularly the use of PSB, have emerged as a promising alternative to conventional chemicals. While previous studies have demonstrated the ability of PSB to enhance rice growth in saline soils, the precise mechanisms underlying this beneficial interaction remain poorly characterized.

Standard cultivation protocols for phototrophic purple non-sulfur bacteria (PNSB) [23,24] are applicable to the isolation of PSB with salt tolerance when supplemented with NaCl. In this study, we successfully isolated and characterized the RH-AZ strain. The bacterial strain intrinsically exhibits a red pigmentation, and this chromogenic compound is not secreted into the extracellular medium (Figure 1A,B). The bacterial cell contains distinctive internal photosynthetic membranes organized as lamellar stacks parallel to the cytoplasmic membrane. Notably, the RH-AZ strain requires NaCl for proliferation, thriving at seawater salinity and moderately higher concentrations, consistent with its classification as a mesophilic marine bacterium [25]. Phylogenetic analysis based on 16S rDNA sequences revealed that the RH-AZ strain was highly distant from *R. palustris*, the type species of the genus *Rhodopseudomonas*. It is closely related to new isolates of the genus Rhodobium from marine environments. This genus belongs to the class *Alphaproteobacteria*, order *Rhizobiales*, and family *Rhodobiaceae* (as per *Bergey’s Manual of Systematics of Archaea and Bacteria*), and currently comprises two recognized species: *R. orientis* and *R. marinum.* Additionally, *R. marinum* displays pronounced cellular motility, with highly active and agile cells observed in culture [25]. This characteristic precisely conforms to our discovery. In the previous study, we isolated another strain RH-I1 (GenBank accession no.PP711549.1) belonging to *R. orientis*. Distinct growth patterns between RH-I1 and RH-AZ emerged during cultivation. During cultivation, RH-I1 exhibited a pronounced tendency to form persistent flocculent suspensions that resisted dispersion through mechanical agitation (unpublished). In contrast, RH-AZ demonstrated distinct phenotypic advantages in liquid culture systems: it displayed vivid pigmentation, maintained homogeneous cell distribution, and showed no flocculent aggregation. Remarkably, even after extended quiescent periods, vigorous agitation readily restored homogeneous distribution throughout the culture medium. Based on the above phylogenetic tree analysis and characterization characteristics, we conducted subsequent experimental analyses using the RH-AZ strain.

Genome mining revealed that the presence of functional genes has become an effective means for efficiently developing and using biocontrol bacterial strains. It provides an in-depth understanding of the genetic and physiological characteristics of strains and has great significance in unraveling the potential of those strains for the production of secondary metabolites, as well as the underlying mechanisms of both plant growth promotion and abiotic stress alleviation [26,27]. Through genomic analysis of *R. marinum* RH-AZ, we found that under saline stress conditions, the RH-AZ strain synthesizes critical ionic regulators of K^+^/Na^+^ homeostasis and key osmoprotectants including glycine betaine, proline, trehalose, and choline. These genetic elements likely underlie the fact that RH-AZ possesses exceptional survival capacity at NaCl concentrations ≤10%. Beyond mitigating osmotic imbalance, solute synthesis protects cellular integrity by stabilizing macromolecular structures and scavenging reactive oxygen species, as demonstrated in conserved stress response pathways [28,29]. The dual mechanisms of ion transport regulation and compatible solute accumulation, a hallmark of bacterial osmoadaptation, enable RH-AZ to simultaneously counter ionic toxicity, oxidative damage, and macromolecular denaturation under hypersaline conditions [30,31].

Detailed genomic characterization of the RH-AZ strain has uncovered an integrated network of functionally coordinated genetic determinants governing plant growth enhancement and environmental adaptation. The organism’s genome encodes comprehensive systems for nitrogen cycle coordination (fixation, assimilation, and metabolism), phytohormone biosynthesis (including auxin pathways), and micronutrient management through siderophore-mediated iron acquisition and phosphate solubilization complexes. Ammonia produced by plant growth-promoting microorganisms supplies nitrogen to host plants, leading to increased plant biomass. This process relies on nitrogen fixation (*nif*) genes, which are required for the maturation of nitrogenase in these bacteria. *Nif* genes are required components of the enzymatic module encoding nitrogenase [32]. The genome of RH-AZ included multiple genes involved in nitrogen fixation and nitrogen metabolism, such as *fixG*, *fixH*, *fixI*, *fixS*, *nifB*, *nifQ*, *nifS*, *nirK*, *gltB* and *gltD*, and some of these genes are involved in ammonia assimilation [33]. Others encode nitrite reductase, which converts nitrate nitrogen into ammonium under aerobic conditions [34]. Auxin is an essential substance for plant growth, and IAA secretion is one of the most important characteristics of some PGPRs [35]. There are five tryptophan-dependent IAA biosynthesis pathways in organisms, including the Indole-3-acetamide, indole-3-acetonitrile, indole-3-pyruvic acid, tryptamine, and tryptophan side-chain oxidase pathways [36]. The genome of RH-AZ included multiple genes involved in auxin biosynthesis, such as *trpA*, *trpB*, *trpC*, *trpD*, and *trpE*. In tryptophan operon, *trpA* and *trpB* catalyze the conversion of indole to tryptophan, while *trpC* catalyzes the cyclization reaction to produce indole-3-glycerol phosphate. Additionally, *trpD* catalyzes the formation of phosphoribosylanthranilate (PRA), and *trpE* catalyzes the reaction between chorismate and glutamine to generate anthranilate, which serves as a precursor for IAA biosynthesis. Another key role of plant growth-promoting bacteria is the production of siderophores [37,38]. The RH-AZ genome encodes a series of siderophore transport-related genes. This capability likely provides RH-AZ with a competitive advantage in diverse environments, enhancing its survival and competitiveness in its ecological niche. Additionally, genomic analysis of the RH-AZ genome revealed genes related to phosphate uptake and transport (*pstS* and *pstB*) and genes involved in regulating phosphate starvation responses (*phoB* and *phoU*). Phosphorus is a crucial nutrient for plants, and its deficiency in available form restricts plant growth and development. The phoBR operon is a regulatory system within the Pho regulon that responds to low phosphate concentrations and *PhoU* plays a crucial role in regulating intracellular phosphate metabolism. The Pst system includes the proteins PstS, PstC, PstA, and PstB [39], which interact with PhoBR, as well as the phosphate modulator PhoU [40]. Particularly noteworthy is the identified acetolactate synthase genes, which suggests novel biosynthetic capabilities in branched-chain amino acid production that may contribute to both microbial survival and plant–microbe interactions. These molecular systems collectively enable RH-AZ to orchestrate three fundamental plant growth-promoting strategies: (1) enhancing nutrient bioavailability through nitrogen provisioning, phosphorus mineralization, and iron chelation; (2) stress mitigation via osmoprotectant synthesis and redox homeostasis mechanisms; and (3) direct plant growth stimulation through phytohormonal regulation. The subsequent experimental results on the bacterial strain promote rice growth under saline stress have confirmed the above conjecture.

In the rice salt tolerance experiment in which RH-AZ participated, the growth characteristics of two rice cultivar plants of 3931 (moderately salt-tolerant) and 9311 (salt-sensitive cultivar), including stem length, root length, fresh stem weight, fresh root weight, stem dry weight, root dry weight, were measured, and the changes in classic salt stress physiological indicators such as chlorophyll, MDA, proline, and antioxidant enzymes of CAT, APX, SOD, and POD were detected. It is well established that salinity significantly inhibits rice plant growth, with substantial variation in salt tolerance observed among different cultivars tested (Figure 4). This growth inhibition primarily stems from salinity-induced reductions in plant water content, which disrupts cellular homeostasis and critical biological processes [41]. Our investigation confirmed these fundamental effects, demonstrating that salt stress not only reduced shoot/root length and fresh weight in rice plants, but also impaired photosynthetic capacity through chlorophyll degradation, ultimately leading to decreased dry biomass accumulation (Figure 5 and Figure 6). Notably, while salinity caused a substantial growth reduction in both rice varieties tested, RH-AZ application significantly mitigated these inhibitory effects, particularly in the T4 and T5 treatments (Figure 4, Figure 5 and Figure 6). Additionally, our results indicated that the application of RH-AZ in the T5 treatment significantly reduced the inhibition of salt stress in rice plants and improved its growth characteristics, mainly through the fermentation metabolites of the RH-AZ strain.

Chlorophyll content serves as a critical biomarker of photosynthetic capacity. Despite significant salt stress-induced degradation of chlorophyll, RH-AZ application effectively reversed this decline (Figure 7A), suggesting that photosynthetic recovery constitutes a key mechanism underlying RH-AZ-mediated salt tolerance. Notably, biomass accumulation exhibited a strong positive correlation with chlorophyll levels across all treatments, consistent with established principles of photosynthesis-growth interdependence under stress [42]. This photosynthetic enhancement likely provides the energy required to sustain salt-tolerance mechanisms. The cellular impacts of salinity were further evidenced by MDA accumulation, a key indicator of membrane damage through lipid peroxidation. When plants are damaged, the disruption of the reactive oxygen metabolism system leads to an increase in the MDA content in lipids, increases membrane permeability, and reduces plant resistance [43,44]. Salt-stressed plants showed elevated MDA levels across T2, T3 and T6 treatments, with the salt-sensitive cultivar 9311 exhibiting particularly pronounced effects. However, RH-AZ inoculation effectively reduced MDA content in both 9391 and 9311 cultivars under T4 and T5 conditions (Figure 7B), demonstrating its protective role in maintaining membrane integrity under saline stress. Plants employ sophisticated osmotic adjustment strategies to counter salt stress, accumulating compatible solutes such as proline, polyols, and various sugars. Our findings aligned with established mechanisms, suggesting that microbial associations can enhance osmolyte production [45,46]. Nguyen et al. [47] indicated that proline has a dual role as an antioxidant and in osmoregulation in salinity stress alleviation. It can effectively reduce the adverse effects of salinity and allow the cells to maintain cellular homeostasis. RH-AZ inoculated rice plants exhibited significantly increased proline levels in two rice cultivars tested in T4 and T5 treatments, suggesting this osmoregulatory compound plays a crucial role in the observed stress mitigation. This inverse relationship between MDA suppression and proline potentiation highlights dual microbial strategies for stress adaptation.

The oxidative stress component of salt toxicity manifested through reactive oxygen species (ROS) accumulation, which at elevated levels causes cellular damage and metabolic disruption [48]. The function of antioxidant enzymatics is to scavenge ROS and free radicals to enhance plant responses against various abiotic stresses [49]. There are numerous instances of microbes enhancing antioxidant enzymatic activity in plants as a crucial method for reducing salt stress [50]. RH-AZ’s protective effects were particularly evident in the activation of antioxidant defenses, with T4 and T5 treatments showing significantly enhanced activities of SOD, APX, CAT, and POD enzymes compared to other treatments. This coordinated enzymatic response facilitates efficient ROS scavenging, maintaining redox homeostasis and minimizing oxidative damage.

In summary, compared to the non-inoculated control subjected to salt stress alone (T2), rice plants inoculated with RH-AZ fermentation broth (T4) or its supernatant (T5) exhibited enhanced growth characteristics, increased antioxidant enzyme activity, elevated proline and chlorophyll content, and reduced malondialdehyde (MDA) levels (Appendix A). In contrast, inoculation with PBLM (T3) or RH-AZ bacterial suspension (T6) showed no significant difference relative to T2. Combined with whole-genome analysis, functional annotation, and secondary metabolite prediction, these results indicate that extracellular metabolites secreted by RH-AZ during fermentation—rather than the bacterial cells themselves—play a pivotal role in mitigating salt stress inhibition in rice plants. Therefore, analysis of RH-AZ extracellular metabolites will be the focus of further research, ultimately aiming for the isolation, characterization, and field application of key bioactive compound(s).

## 4. Materials and Methods

### 4.1. Microbial Cultivation and Salt Stress Tolerance Assay

The bacterial strain *R. marinum* RH-AZ was isolated from seawater collected at Laoshan Mountain, Qingdao City, Shandong Province, China. Routine cultivation was performed at 32 °C on double-layer solid medium containing 3% (*w*/*v*) NaCl under continuous illumination with light intensity of 3000 lux. Single colonies were subcultured, and pure cultures of RH-AZ were grown in a photosynthetic bacterial liquid medium (PBLM) supplemented with 3% (*w*/*v*) NaCl to prepare bacterial fermentation broth. Strain morphology was analyzed using field-emission SEM and TEM (Hitachi High-Tech Co., Tokyo, Japan).

To evaluate salt stress tolerance, the liquid medium supplemented with varying NaCl concentrations from 1% to 10% (*w*/*v*) was used. Cultures were incubated at 32 °C under continuous illumination (3000 lux) for 45 days. Optical density at 600 nm (OD600) was measured daily for each treatment group.

### 4.2. Phylogenetic Analysis

Genomic DNA was extracted from the RH-AZ strain using the CTAB method [51]. Subsequently, the 16S rDNA of RH-AZ was amplified by PCR using the universal primers 27F (5′-AGAGTTTGATCCTGGCTCAG-3′) and 1492R (5′-GGTTACCTTGTTACGACTT-3′). The PCR reaction included 1 μL of KOD- Plus-Neo (Toyobo Life Scientific, Shanghai, China), 5 μL of 10 × PCR buffer for KOD-Plus-Neo, 3 μL of 25 mmol/L MgSO_4_, 5 μL of 2 mmol/L dNTPs, 1 μL of each primer (10 mmol/L), 1 μL of DNA template, and 33 μL of ultrapure water, in a total volume of 50 μL. The following cycling conditions were used for PCR: 94 °C for 2 min, 35 cycles of 15 s at 98 °C, 15 s at 58 °C, and 90 s at 68 °C. The PCR products were purified with an EasyPure PCR Purification Kit (TransGene Biotech, Beijing, China) and then sequenced. A phylogenetic tree of the RH-AZ strain was constructed using 16S rDNA gene sequences of another 16 strains of PSB retrieved from the NCBI database. The evolutionary history was inferred using the Neighbor-Joining method. The optimal tree is shown. The percentage of replicate trees in which the associated taxa clustered together in the bootstrap test (1000 replicates) are shown next to the branches. The tree is drawn to scale, with branch lengths in the same units as those of the evolutionary distances used to infer the phylogenetic tree. The evolutionary distances were computed using the Maximum Composite Likelihood method and are in the units of the number of base substitutions per site. Evolutionary analyses were conducted in Molecular Evolutionary Genetics Analysis (MEGA) software version 11 (Institute for Genomics and Evolutionary Medicine, Temple University, Philadelphia, PA, USA) [52].

### 4.3. Genome Sequencing of the RH-AZ Strain

#### 4.3.1. Genomic DNA Extraction

Whole-genome shotgun DNA sequencing was performed by OEbiotech (Shanghai, China) using paired-end NovaSeq 6000 (Illumina, San Diego, CA, USA) and long-read PacBio RS (Pacific Biosciences, Menlo Park, CA, USA) platforms. Bacterial cells were pelleted via centrifugation, and the supernatant was discarded. High-quality genomic DNA from the RH-AZ strain was extracted using an optimized CTAB method. DNA concentration and purity were assessed with a NanoDrop 2000 spectrophotometer (Thermo Fisher Scientific, Waltham, MA, USA), while quantification was performed using a Qubit 3.0 fluorometer (Life Technologies, Carlsbad, CA, USA). DNA integrity was verified by 0.8% agarose gel electrophoresis.

#### 4.3.2. Whole-Genome Sequencing and Functional Annotation

Upon quality validation, a portion of the DNA was randomly sheared using a Covaris ultrasonicator for Illumina Library Preparation. Library construction involved end repair, A-tailing, adapter ligation, size selection with magnetic beads, PCR amplification, and bead-based purification. Libraries were preliminary quantified with Qubit 3.0, diluted to appropriate concentrations, and assessed for insert size distribution using an Agilent 2100 Bioanalyzer (Agilent, Santa Clara, CA, USA). Qualified libraries were pooled based on effective concentrations and sequenced on the Illumina NovaSeq 6000 platform after clustering on a cBOT instrument.

Another aliquot of DNA was fragmented to ~10 kb using a Covaris g-Tube, followed by purification with 0.45× magnetic beads to prepare PacBio Library. A total of 2.5 μg DNA was used for SMRTbell library construction. The workflow included exonuclease digestion to remove single-stranded overhangs, damage repair, blunt-end repair, ligation of barcoded SMRTbell adapters, and exonuclease treatment to remove unligated fragments. Final libraries were purified with 0.45 × PB magnetic beads and validated using the same quality control methods as aforementioned. Sequencing was performed on the PacBio Sequel II platform (targeting specified data yields), followed by data processing with SMRT Link 10.1.0 software. PacBio reads were assembled using a hybrid approach with SMRT Link 10.1 (microbial assembly module), HGAP4, and Canu (v1.6). Protein-coding genes were predicted using Glimmer (v3.02). A circular genome map of the RH-AZ strain was generated with Circos (v0.64).

### 4.4. Genome Characteristics of the RH-AZ Strain

TRNA and rRNA genes were identified using tRNAscan-SE (v2.0) and RNAmmer (v1.2), respectively. Noncoding RNAs excluding tRNA and rRNA were predicted through the cmscan program in Infernal (v1.1.4). TRF software and MISA2.1 were used to predict tandem repeats (TRs) and simple sequence repeats (SSR), respectively. Genomic CRISPRs prediction was performed using the Mining CRISPRs in Environmental Datasets (MinCED), Genomics Islands (GIs) prediction was performed using IslandPath-1.0.6, and Prophage was predicted by PhiSpy v4.2.19.

### 4.5. Genomic Analysis and Functional Annotation

Protein functional annotation of the RH-AZ genome was performed using the Clusters of Orthologous Groups (COG) database. The DIAMOND software (v2.0.9.147) BLASTP algorithm was employed to align predicted protein sequences against the COG-2020 database with an E-value threshold of 10^−5^, where the highest-scoring alignment was selected for final annotation. Secondary metabolite biosynthetic gene clusters were identified using antiSMASH (version 7.1.0).

### 4.6. Plant Materials and Growth Conditions

Two rice cultivars, 3931 and 9311 (Appendix A), maintained in our laboratory were used as for moderately salt-tolerant and salt-sensitive cultivars, respectively. Seeds were surface-sterilized with 3% (*v*/*v*) sodium hypochlorite (NaClO) for 20 min, followed by three rinses with 250 mL of sterile distilled water per rinse. Sterilized seeds were transferred to a sterile beaker and soaked in distilled water until radicle emergence approximately 2–3 mm. During this period, seeds were washed and the distilled water was replaced daily. Excess water was drained, and seeds were maintained under high humidity by covering them with moist sterile filter paper to promote uniform germination. After 1–2 days, germinated seeds were transferred to 96-well hydroponic culture boxes filled with Hogland nutrient solution (LB0140; Solarbio Science and Technology Co., Ltd., Beijing, China). Hydroponic seedlings were grown in a controlled-environment greenhouse (28 °C, 70% relative humidity (RH), a 12:12 h light/dark photoperiod) and subjected to experimental treatments at the three-leaf stage.

### 4.7. Plant Salt Stress and Treatment Applications

PBLM containing 3% (*w*/*v*) NaCl was prepared and sterilized by autoclaving. Half of the total PBLM volume was inoculated with the RH-AZ strain and incubated to prepare the RH-AZ fermentation broth. Half of the RH-AZ fermentation broth was centrifuged (6000× *g*, 10 min) to separate the cells from the supernatant. The RH-AZ supernatant was filter-sterilized and stored for subsequent use. The resulting cell pellet was resuspended in an equal volume of sterile PBLM to prepare the RH-AZ bacterial suspension, which was also stored for subsequent use. Uniformly grown rice seedlings were assigned to six treatments: T1: Negative control (Hoagland nutrient solution), T2: Non-inoculated control (Hoagland nutrient solution + NaCl), T3: Hoagland nutrient solution + NaCl + PBLM, T4: Hoagland nutrient solution + NaCl + RH-AZ fermentation broth, T5: Hoagland nutrient solution + NaCl + centrifuged RH-AZ supernatant (sterile-filtered), T6: Hoagland nutrient solution + NaCl + RH-AZ bacterial suspension. The final NaCl concentration was adjusted to 150 mM for the moderately salt-tolerant cultivar 3931 and 100 mM for the salt-sensitive cultivar 9311.

All treatments were applied to both rice cultivars including 3931 and 9311 in triplicate. After 14 days of continuous exposure to different treatments, plants were transferred to fresh Hoagland nutrient solution for two weeks to assess rice recovery. Hydroponic solutions were replaced twice weekly to maintain nutrient availability for plant growth. Experiments were conducted in a controlled greenhouse (28 °C, 70% RH, a 12 h light/dark photoperiod).

### 4.8. Effect of RH-AZ Inoculation on Rice Growth Parameters Under Salt Stress

Plant growth parameters were evaluated at 0, 3, 6 and 9 days post-salt stress induction. Five vigorous seedlings per treatment group were selected for analysis. Following harvest, roots were gently rinsed with deionized (Dl) water to eliminate residual hydroponic growth substrate. Shoots and roots were sectioned at the crown junction. The following measurements were conducted: (1) Shoot and root length: Measured using digital calipers; (2) Fresh biomass: Separately weighed for shoots and roots immediately; (3) Dry biomass: Tissues were oven-dried at 70 °C for 72 h until constant weight. The dry weight was recorded after drying of the samples. All measurements were performed in quintuplicate under controlled laboratory conditions (25 °C, 60% RH).

### 4.9. Analysis of Chlorophyll, Malondialdehyde, Proline, and Antioxidant Enzyme Activities

Chlorophyll quantification followed the protocol of Witham et al. [53]. Fresh leaf tissue (0.1 g) harvested from each treatment group at four intervals (6, 9, 12, 15 days) post-stress was homogenized in 80% ice-cold acetone and brought to a final volume of 10 mL with additional chilled acetone. Absorbance measurements at 452, 663, and 645 nm were conducted using a UV-EON spectrophotometer (BioTek, Kyoto, Japan). Total chlorophyll content was calculated according to the equations provided by Sadashivam and Manickam [54].

Lipid peroxidation was quantified through MDA accumulation using the thiobarbituric acid reactivity assay. Intracellular ROS levels were indirectly assessed by profiling antioxidant enzyme activities, including CAT, APX, POD, and SOD. Vigorous seedlings were harvested from each treatment group at three intervals (3, 6, 9 days) post-stress. Leaf tissues were aseptically excised into 0.1 g aliquots, transferred to pre-labeled 2 mL cryovials, flash-frozen in liquid N_2_, and stored at −80 °C until biochemical analysis. MDA levels and proline contents, and antioxidant enzyme (CAT, APX, POD and SOD) activities were analyzed using Solarbio microassay kits (Solarbio Science and Technology Co., Ltd., Beijing, China). For MDA quantification, absorbance readings at 532, 600, and 450 nm were obtained using the UV-EON spectrophotometer, with calculations incorporating correction factors for nonspecific turbidity. Proline content was determined at 520 nm using a standard curve as described by Sadashivam and Manickam [54]. Antioxidant enzyme activities were assessed by monitoring absorbance changes at enzyme-specific wavelengths: 240 nm (CAT), 290 nm (APX), 436 nm (POD), and 560 nm (SOD), with calculations based on established molar extinction coefficients.

### 4.10. Data Analysis

All measurements were replicated at least three times. Statistical analyses were performed using SPSS version 24 (IBM SPSS, Chicago, IL, USA) to evaluate the individual and interactive effects of salt concentration, bacterial treatment, and time duration on various parameters including plant growth characteristics, chlorophyll content, MDA levels, Proline content, and antioxidant enzyme (CTA, PAX, POD, SOD) activities. Experimental data visualizations were created using SigmaPlot version 14.0 (Systat Software Inc., San Jose, CA, USA). Statistical significance was determined at *p* < 0.05 for all comparisons.

## 5. Conclusions

In this study, we identified and analyzed the entire genome of the bacteria of a RH-AZ strain of PSB. We analyzed the functions of related genes related to salt tolerance characteristics in the RH-AZ strain and its growth-promoting effects on rice plants. The multilayered genetic architecture and rice salt tolerance assays provided a mechanistic foundation for the efficacy of the RH-AZ strain in enhancing crop growth under saline stress conditions through simultaneous nutritional support and stress-priming activation. The RH-AZ strain affected key stress mitigation biomarkers: it enhanced the activity of antioxidant enzymes including SOD, POD, CAT, and APX, increased the contents of proline and chlorophyll in plants, and reduced the content of MDA. These results demonstrate that RH-AZ inoculation enhances the salt tolerance of rice through multiple complementary mechanisms: maintaining membrane stability, optimizing osmotic balance, preserving photosynthetic capacity, and strengthening antioxidant defenses. Additionally, our results indicate that the application of the RH-AZ strain significantly reduced the inhibition of salt stress in rice and improved the growth characteristics of rice mainly through its fermentation metabolites. These findings present a promising microbial approach for rice cultivation in coastal and saline-affected farmlands. Further investigation of RH-AZ metabolites could make critical contributions to enhancing global rice production and mitigating food insecurity.

## Figures and Tables

**Figure 1 plants-14-02516-f001:**
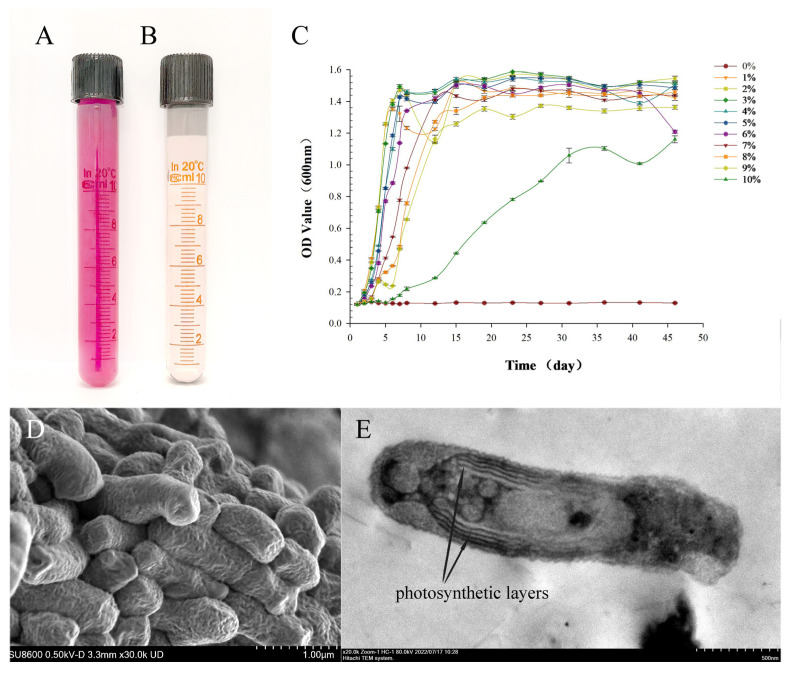
Macroscopic, microscopic, and growth kinetics characterization of RH-AZ. (**A**) Phototrophic liquid cultures of RH-AZ; (**B**) fermentation supernatant of RH-AZ; (**C**) the growth of RH-AZ under varying NaCl concentrations of 1–10% (*w*/*v*); strain morphology was analyzed using field-emission scanning electron microscopy (**D**) and transmission electron microscopy (**E**).

**Figure 2 plants-14-02516-f002:**
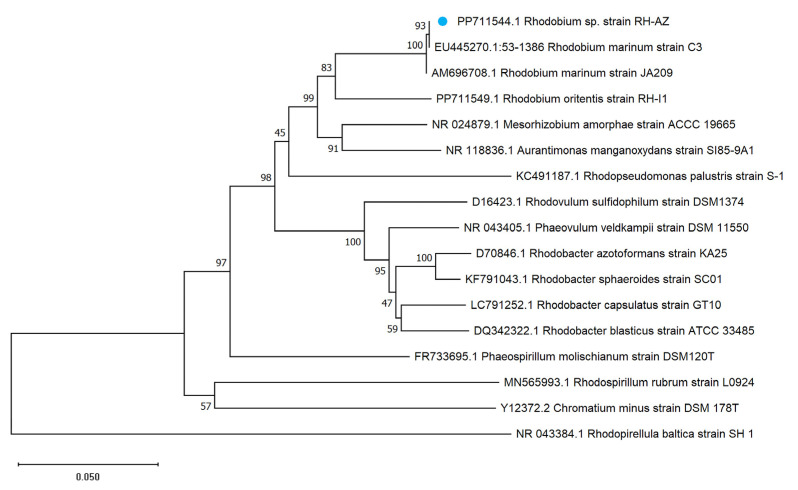
Phylogenetic classification of RH-AZ. Blue dots denote the position of RH-AZ. *Rhodopirellula baltica* strain SH 1 was included as an outgroup.

**Figure 3 plants-14-02516-f003:**
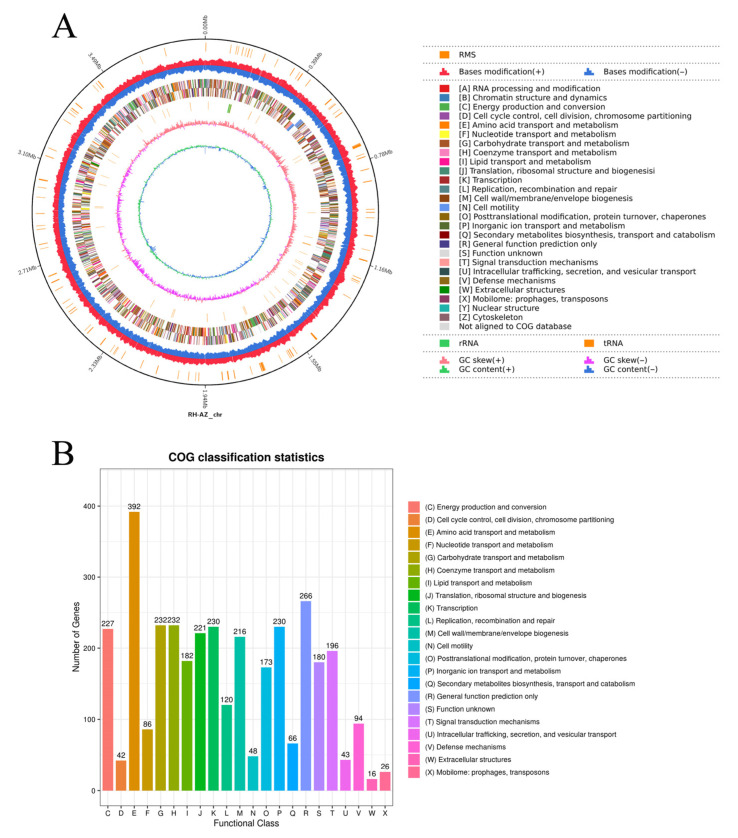
Circular representation of the RH-AZ genome structure (**A**) and functional comparisons among RH-AZ were provided by the COG database (**B**).

**Figure 4 plants-14-02516-f004:**
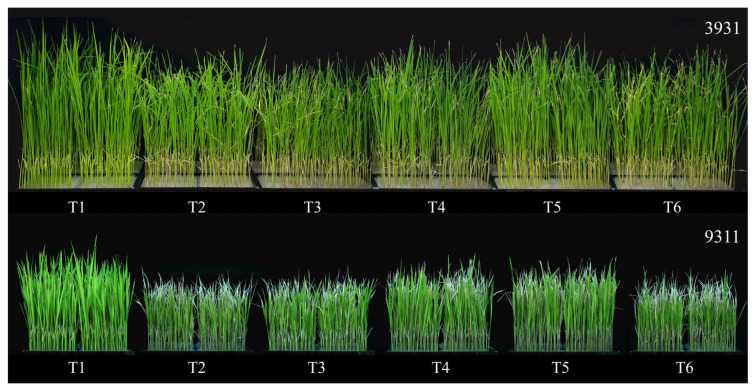
The growth potential of moderately salt-tolerant cultivar 3931 and salt-sensitive cultivar 9311 under different experimental treatments. T1: Negative control (Hoagland nutrient solution), T2: non-inoculated control (Hoagland nutrient solution + NaCl), T3: Hoagland nutrient solution + NaCl + photosynthetic bacterial liquid medium (PBLM), T4: Hoagland nutrient solution + NaCl + RH-AZ fermentation broth, T5: Hoagland nutrient solution + NaCl + centrifuged RH-AZ supernatant (sterile-filtered), T6: Hoagland nutrient solution + NaCl + RH-AZ bacterial suspension. (The final NaCl concentration was adjusted to 150 mM for the moderately salt-tolerant cultivar 3931 and 100 mM for the salt-sensitive cultivar 9311).

**Figure 5 plants-14-02516-f005:**
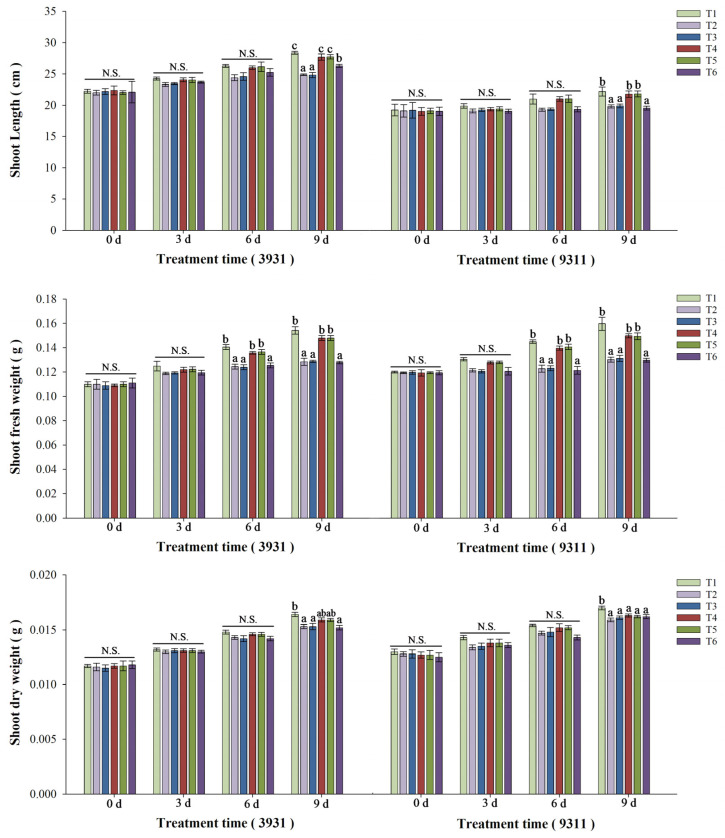
The growth characteristics of the shoots of moderately salt-tolerant cultivar 3931 and salt-sensitive cultivar 9311 under different experimental treatments. The T1–T6 experimental treatments are the same as those shown in Figure 4. Different letters above each bar indicate statistically significant differences (P < 0.05) within the same cultivar at a given time point by Duncan’s multiple range test; N.S. denotes non-significance. Error bars indicate standard deviation.

**Figure 6 plants-14-02516-f006:**
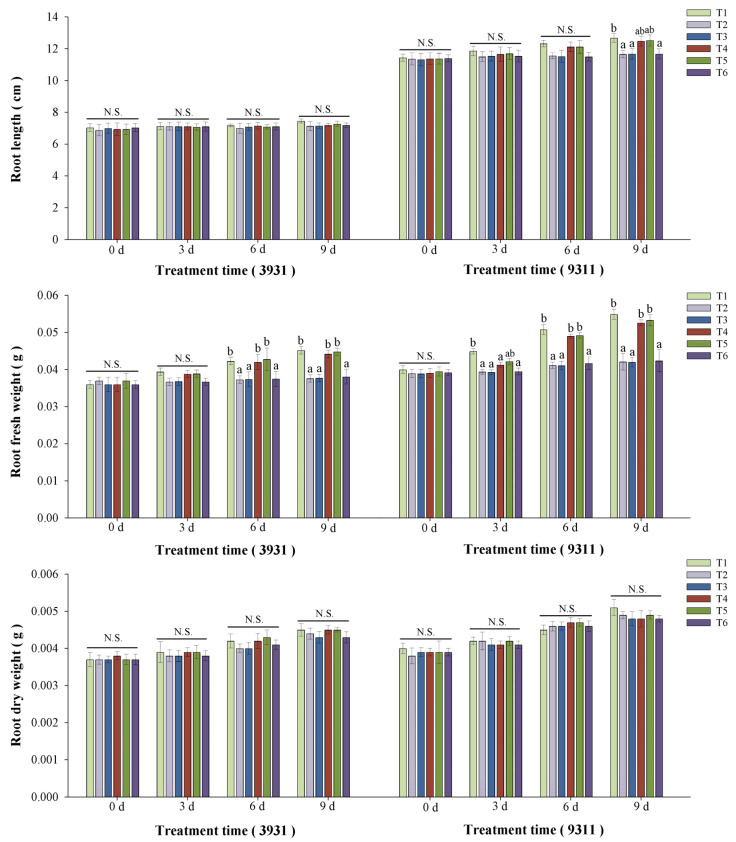
The growth characteristics of the roots of moderately salt-tolerant cultivar 3931 and salt-sensitive cultivar 9311 under different experimental treatments. The T1–T6 experimental treatments are the same as those shown in Figure 4. Different letters above each bar indicate statistically significant differences (P < 0.05) within the same cultivar at a given time point by Duncan’s multiple range test; N.S. denotes non-significance. Error bars indicate standard deviation.

**Figure 7 plants-14-02516-f007:**
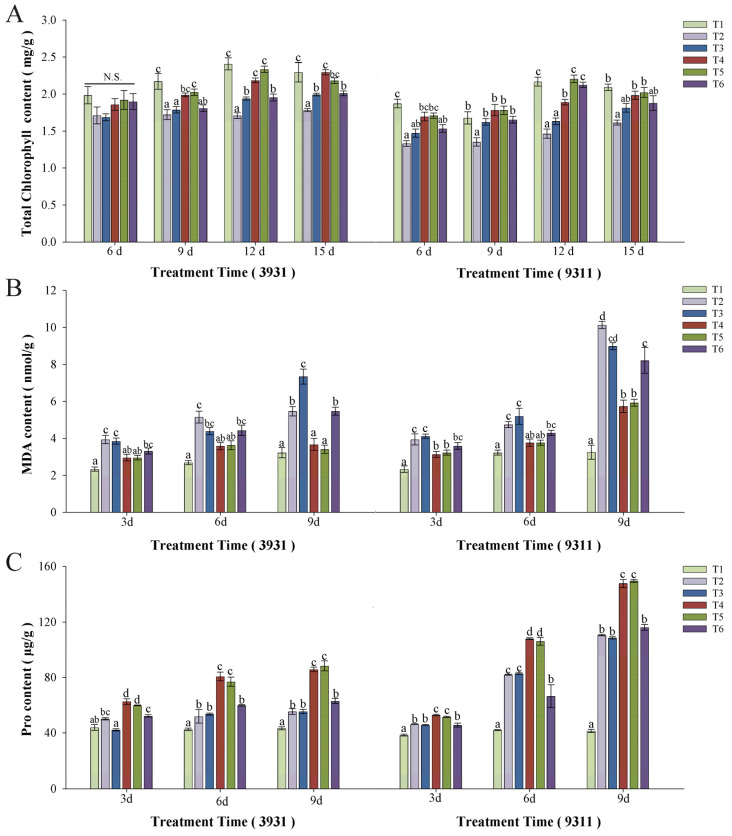
Effect of inoculation of RH-AZ on (**A**) chlorophyll content; (**B**) MDA content, and (**C**) proline content. Different letters above each bar indicate statistically significant differences (P < 0.05) within the same cultivar at a given time point by Duncan’s multiple range test; N.S. denotes non-significance. Error bars indicate standard deviation.

**Figure 8 plants-14-02516-f008:**
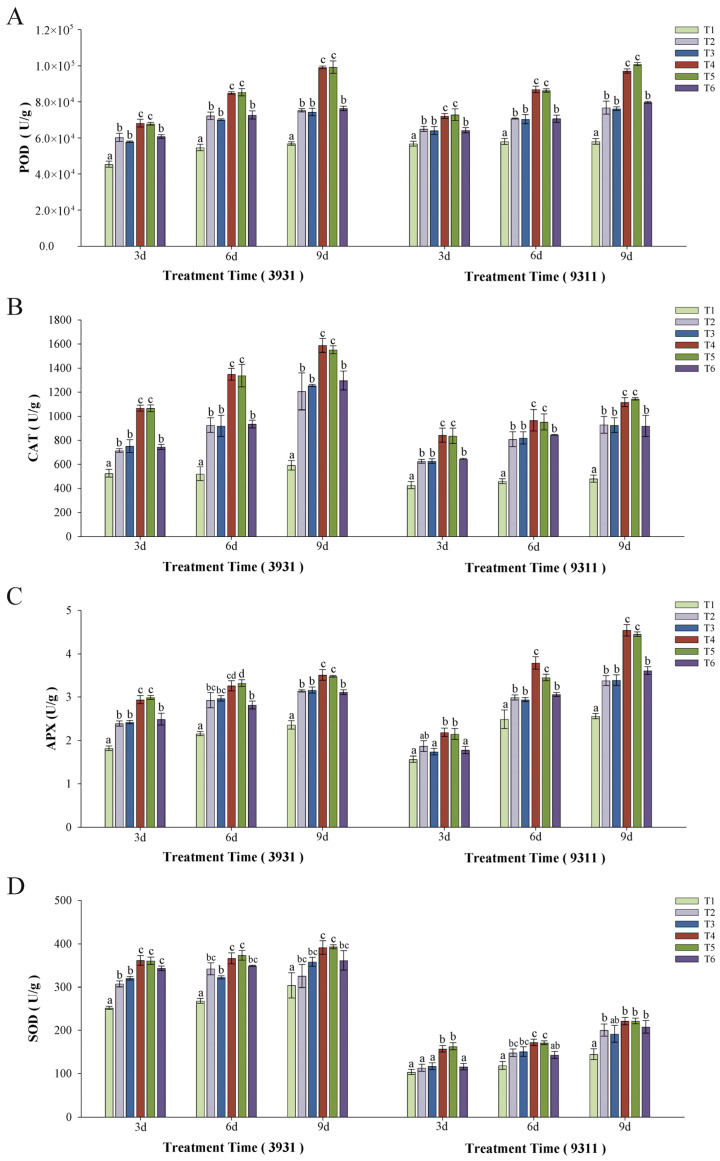
Effect of inoculation of RH-AZ on the antioxidant enzyme activity in two rice varieties, moderately salt-tolerant cultivar 3931 and salt-sensitive cultivar 9311. (**A**) Peroxidase (POD), (**B**) catalase (CAT), (**C**) ascorbate peroxidase (APX), and (**D**) superoxide dismutase (SOD). Different letters above each bar indicate statistically significant differences (P < 0.05) within the same cultivar at a given time point by Duncan’s multiple range test. Error bars indicate standard deviation.

## Data Availability

The data supporting this study’s findings are available from the corresponding authors, J.P. and D.Z., upon reasonable request.

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
