# Peer review of "Functional Analyses of a Rhodobium marinum RH-AZ Genome and Its Application for Promoting the Growth of Rice Under Saline Stress"

_plants, 2025, doi:10.3390/plants14162516_

Round 1
Reviewer 1 Report
Comments and Suggestions for Authors
The article by the authors Yang Gao , Cheng Xu , Tao Tang , Xiao Xie , Renyan Huang , Youlun Xiao , Xiaobin Shi , Huiying Hu , Yong Liu , Jing Peng, and Deyong Zhang « Functional Analyses of a Rhodobium marinum RH-AZ Genome and Its Application for Promoting the Growth of Rice Under Saline Stress » touches upon one of the most important problems in the production of agricultural crops - salt tolerance.
The problematic is described well in the introduction.
However, it is necessary to write about the use of microorganisms in general in agriculture to improve the quality of grain and green mass, since this topic is not new.
Тhen necessary to clarify whether other microorganisms are used to address the problem of plant resistance to salts in agricultural soils.
It should be clarified that the choice of these particular microorganisms is apparently associated with the possibility of fixing salts, which is due to their natural habitat.
The last paragraph of the introduction needs to be made more structured and logical.
It should be noted that inoculation with microorganisms is one of the ways to solve the problems of salt tolerance.
List the main ions that cause negative effects on plants when grown on saline soils.
Line 90 decipher CFU
Line 85-86 color of bacteria colonies grown on agar indicates the accumulation of pigment.
What does fermentation indicate?
Line 94-97 shows a morphology typical for this type of bacteria?
Note the morphological features in the figure. A reference confirming the species morphology is also required.
Line 145-148, decipher the abbreviated names of the biosynthesis of secondary metabolites.
Line 152-153 describes the genetically determined response to elevated salt concentrations with or without inoculation?
Line 149-164 The authors describe two lines of rice: salt-resistant and salt-sensitive, plus with and without inoculation. Due to the large number of experiments, it is better to note this in the text to avoid confusion and for better understanding.
Line 187 proline synthesis were 2.0–3.6 187 fold (P < 0.05) less or more compared with T1 control?
In the "Discussion" section, you compare 2 bacterial lines. One you are studying in the current study, the other previously. To what extent were the conditions of colony growth and their study identical? Are such data comparable?
Before the results section, it is better to say that you are characterizing a bacterial line to clarify or suggest a mechanism for mitigating the effects of salts on plants or specifically on rice, to identify the potential of a specific bacterial species to improve salt tolerance in important agricultural crops.
Explain the choice of indicators for confirming salt tolerance (Chlorophyll, Malondialdehyde, Proline, and Antioxidant Enzyme Activity)
The abstract must be corrected in accordance with the corrected text of the article.
Reviewer 2 Report
Comments and Suggestions for Authors
The manuscript demonstrate that R. marinum RH-AZ enhances rice salt tolerance through genomic-encoded adaptive mechanisms and physiological regulation. Therefore, I suggest accept after minor revisions.
The minor revisions please refer to the attachment.

Reviewer 3 Report
Comments and Suggestions for Authors
Manuscript plants-3754772 “Functional Analyses of a Rhodobium marinum RH-AZ Genome and Its Application for Promoting the Growth of Rice Under 3 Saline Stress” is an interesting paper bringing some valuable and new information about microorganism interaction in rice for saline tolerance.
Overall, writing is clear and experiments are sound. However instead of the good analysis performed, the present version of the manuscript presents some deficiencies which must be revised before publication. Manuscript is confused in the description of the material assayed and the methodology. Discussion of results and Conclusions should also be improved.
Objectives of the work must be clarified indicating the main novelty of the work.
Quality of Figure 4 should be improved increasing font size.
In the discussion, section, a hypothetical scheme showing the molecular effect of RH-AZ.
The conclusions section must be added as a separated section indicating the main implications of the obtained results for rice production and breeding.
In Plant Materials, authors must describe main characteristics of the assayed rice genotype. This information should be added as supplementary material.
Experimental analysis must also be clarified with an accurate description of treatment and phenological analysis. Authors must clarify the relation between the RH molecular characterization and phylogeny and the interaction with the rice to increase salt tolerance.
Round 2
Reviewer 1 Report
Comments and Suggestions for Authors
The authors have taken all the comments into account and revised the text to address all the comments. They have provided comprehensive explanations for all the comments. In its current form, the article can be accepted for publication.
Author Response
Thank you for your notification regarding the acceptance of our manuscript entitled "Functional analyses of a Rhodobium marinum RH-AZ genome and its application for promoting the growth of rice under saline stress". We are pleased to confirm that:
1.Final manuscript has been prepared according to the journal's formatting guidelines
2.Proofreading of all textual and graphical elements has been conducted
We appreciate you efficient handling of our work and look forward to the publication process.
Reviewer 3 Report
Comments and Suggestions for Authors
Authors have revised correctly the manuscript
Author Response

(The authors gave the same response as above.)
